# Using Social Dynamics to Make Individual Predictions: Variational Inference with a Stochastic Kinetic Model

**Zhen Xu, Wen Dong, and Sargur Srihari**
Department of Computer Science and Engineering
University at Buffalo
`{zxu8,wendong,srihari}@buffalo.edu`

## Abstract

Social dynamics is concerned primarily with interactions among individuals and the resulting group behaviors, modeling the temporal evolution of social systems via the interactions of individuals within these systems. In particular, the availability of large-scale data from social networks and sensor networks offers an unprecedented opportunity to predict state-changing events at the individual level. Examples of such events include disease transmission, opinion transition in elections, and rumor propagation. Unlike previous research focusing on the collective effects of social systems, this study makes efficient inferences at the individual level. In order to cope with dynamic interactions among a large number of individuals, we introduce the stochastic kinetic model to capture adaptive transition probabilities and propose an efficient variational inference algorithm the complexity of which grows *linearly* — rather than exponentially— with the number of individuals. To validate this method, we have performed epidemic-dynamics experiments on wireless sensor network data collected from more than ten thousand people over three years. The proposed algorithm was used to track disease transmission and predict the probability of infection for each individual. Our results demonstrate that this method is more efficient than sampling while nonetheless achieving high accuracy.

## 1   Introduction

The field of social dynamics is concerned primarily with interactions among individuals and the resulting group behaviors. Research in social dynamics models the temporal evolution of social systems via the interactions of the individuals within these systems [9]. For example, opinion dynamics can model the opinion state transitions of an entire population in an election scenario [3], and epidemic dynamics can predict disease outbreaks ahead of time [10]. While traditional social-dynamics models focus primarily on the macroscopic effects of social systems, often we instead wish to know the answers to more specific questions. Given the movement and behavior history of a subject with Ebola, can we tell how many people should be tested or quarantined? City-size quarantine is not necessary, but family-size quarantine is insufficient. We aim to model a method to evaluate the paths of illness transmission and the risks of infection for *individuals*, so that limited medical resources can be most efficiently distributed.

The rapid growth of both social networks and sensor networks offers an unprecedented opportunity to collect abundant data at the individual level. From these data we can extract temporal interactions among individuals, such as meeting or taking the same class. To take advantage of this opportunity, we model social dynamics from an individual perspective. Although such an approach has considerable potential, in practice it is difficult to model the dynamic interactions and handle the costly computations when a large number of individuals are involved. In this paper, we introduce an

event-based model into social systems to characterize their temporal evolutions and make tractable inferences on the individual level.

Our research on the temporal evolutions of social systems is related to dynamic Bayesian networks and continuous time Bayesian networks [13, 18, 21]. Traditionally, a coupled hidden Markov model is used to capture the interactions of components in a system [2], but this model does not consider dynamic interactions. However, a stochastic kinetic model is capable of successfully describing the interactions of molecules (such as collisions) in chemical reactions [12, 22], and is widely used in many fields such as chemistry and cell biology [1, 11]. We introduce this model into social dynamics and use it to focus on individual behaviors.

A challenge in capturing the interactions of individuals is that in social dynamics the state space grows exponentially with the number of individuals, which makes exact inference intractable. To resolve this we must apply approximate inference methods. One class of these involves sampling-based methods. Rao and Teh introduce a Gibbs sampler based on local updates [20], while Murphy and Russell introduce Rao-Blackwellized particle filtering for dynamic Bayesian networks [17]. However, sampling-based methods sometimes mix slowly and require a large number of samples/particles. To demonstrate this issue, we offer empirical comparisons with two major sampling methods in Section 4. An alternative class of approximations is based on variational inference. Opper and Sanguinetti apply the variational mean field approach to factor a Markov jump process [19], and Cohn and El-Hay further improve its efficiency by exploiting the structure of the target network [4]. A problem is that in an event-based model such as a stochastic kinetic model (SKM), the variational mean field is not applicable when a single event changes the states of two individuals simultaneously. Here, we use a general expectation propagation principle [14] to design our algorithm.

This paper makes three contributions: First, we introduce the discrete event model into social dynamics and make tractable inferences on both individual behaviors and collective effects. To this end, we apply the stochastic kinetic model to define adaptive transition probabilities that characterize the dynamic interaction patterns in social systems. Second, we design an efficient variational inference algorithm whose computation complexity grows linearly with the number of individuals. As a result, it scales very well in large social systems. Third, we conduct experiments on epidemic dynamics to demonstrate that our algorithm can track the transmission of epidemics and predict the probability of infection for each individual. Further, we demonstrate that the proposed method is more efficient than sampling while nonetheless achieving high accuracy.

The remainder of this paper is organized as follows. In Section 2, we briefly review the coupled hidden Markov model and the stochastic kinetic model. In Section 3, we propose applying a variational algorithm with the stochastic kinetic model to make tractable inferences in social dynamics. In Section 4, we detail empirical results from applying the proposed algorithm to our epidemic data along with the proximity data collected from sensor networks. Section 5 concludes.

## 2  Background

### 2.1  Coupled Hidden Markov Model

A coupled hidden Markov model (CHMM) captures the dynamics of a discrete time Markov process that joins a number of distinct hidden Markov models (HMMs), as shown in Figure 2.1(a). $\mathbf{x}_t = (x_t^{(1)}, \ldots, x_t^{(M)})$ defines the hidden states of all HMMs at time $t$, and $x_t^{(m)}$ is the hidden state of HMM $m$ at time $t$. $\mathbf{y}_t = (y_t^{(1)}, \ldots, y_t^{(M)})$ are observations of all HMMs at time $t$, and $y_t^{(m)}$ is the observation of HMM $m$ at time $t$. $P(\mathbf{x}_t|\mathbf{x}_{t-1})$ are transition probabilities, and $P(\mathbf{y}_t|\mathbf{x}_t)$ are emission probabilities for CHMM. Given hidden states, all observations are independent. As such, $P(\mathbf{y}_t|\mathbf{x}_t) = \prod_m P(y_t^{(m)}|x_t^{(m)})$, where $P(y_t^{(m)}|x_t^{(m)})$ is the emission probability for HMM $m$ at time $t$. The joint probability of CHMM can be defined as follows:

$$P\left(\mathbf{x}_{1,\ldots,T}, \mathbf{y}_{1,\ldots,T}\right) = \prod_{t=1}^{T} P(\mathbf{x}_t|\mathbf{x}_{t-1})P(\mathbf{y}_t|\mathbf{x}_t). \tag{1}$$

For a CHMM that contains $M$ HMMs in a binary state, the state space is $2^M$, and the state transition kernel is a $2^M \times 2^M$ matrix. In order to make exact inferences, the classic forward-backward algorithm sweeps a forward/filtering pass to compute the forward statistics $\alpha_t(\mathbf{x}_t) = P(\mathbf{x}_t|\mathbf{y}_{1,\ldots,t})$

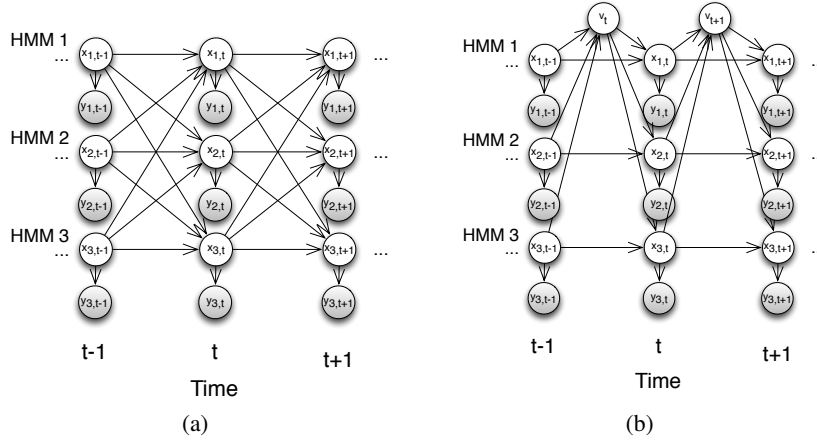

Figure 1: Illustration of (a) Coupled Hidden Markov Model, (b) Stochastic Kinetic Model.

and a backward/smoothing pass to estimate the backward statistics $\beta_t(\mathbf{x}_t) = \frac{P(\mathbf{y}_{t+1,\ldots,T}|\mathbf{x}_t)}{P(\mathbf{y}_{t+1,\ldots,T}|\mathbf{y}_{1,\ldots,t})}$. Then it can estimate the one-slice statistics $\gamma_t(\mathbf{x}_t) = P(\mathbf{x}_t|\mathbf{y}_{1,\ldots,T}) = \alpha_t(\mathbf{x}_t)\beta_t(\mathbf{x}_t)$ and two-slice statistics $\xi_t(\mathbf{x}_{t-1}, \mathbf{x}_t) = P(\mathbf{x}_{t-1}, \mathbf{x}_t|\mathbf{y}_{1,\ldots,T}) = \frac{\alpha_{t-1}(\mathbf{x}_{t-1})P(\mathbf{x}_t|\mathbf{x}_{t-1})P(\mathbf{y}_t|\mathbf{x}_t)\beta_t(\mathbf{x}_t)}{P(\mathbf{y}_t|\mathbf{y}_{1,\ldots,t-1})}$. Its complexity grows exponentially with the number of HMM chains. In order to make tractable inferences, certain factorizations and approximations must be applied. In the next section, we introduce a stochastic kinetic model to lower the dimensionality of transition probabilities.

## 2.2 The Stochastic Kinetic Model

A stochastic kinetic model describes the temporal evolution of a chemical system with $M$ species $\mathcal{X} = \{X_1, X_2, \cdots, X_M\}$ driven by $V$ events (or chemical reactions) parameterized by rate constants $\mathbf{c} = (c_1, \ldots, c_V)$. An event (chemical reaction) $k$ has a general form as follows:

$$r_1 X_1 + \cdots + r_M X_M \xrightarrow{c_k} p_1 X_1 + \cdots + p_M X_M.$$

The species on the left are called *reactants*, and $r_m$ is the number of $m$th reactant molecules consumed during the reaction. The species on the right are called *products*, and $p_m$ is the number of $m$th product molecules produced in the reaction. Species involved in the reaction ($r_m > 0$) without consumption or production ($r_m = p_m$) are called *catalysts*. At any specific time $t$, the populations of the species is $\mathbf{x_t} = (x_t^{(1)}, \ldots, x_t^{(M)})$. An event $k$ happens with rate $h_k(\mathbf{x_t}, c_k)$, determined by the rate constant and the current population state [22]:

$$h_k(\mathbf{x_t}, c_k) = c_k g_k(\mathbf{x_t}) = c_k \prod_{m=1}^{M} g_k^{(m)}(x_t^{(m)}). \tag{2}$$

The form of $g_k(\mathbf{x_t})$ depends on the reaction. In our case, we adopt the product form $\prod_{m=1}^{M} g_k^{(m)}(x_t^{(m)})$, which represents the total number of ways that reactant molecules can be selected to trigger event $k$ [22]. Event $k$ changes the populations by $\mathbf{\Delta_k} = \mathbf{x}_t - \mathbf{x}_{t-1}$. The probability that event $k$ will occur during time interval $(t, t + dt]$ is $h_k(\mathbf{x_t}, c_k)dt$. We assume at each discrete time step that no more than one event will occur. This assumption follows the linearization principle in the literature [18], and is valid when the discrete time step is small. We treat each discrete time step as a unit of time, so that $h_k(\mathbf{x_t}, c_k)$ represents the probability of an event.

In epidemic modeling, for example, an infection event $v_i$ has the form $S + I \xrightarrow{c_i} 2I$, such that a susceptible individual ($S$) is infected by an infectious individual ($I$) with rate constant $c_i$. If there is only one susceptible individual (type $m = 1$) and one infectious individual (type $m = 2$) involved in this event, $h_i(\mathbf{x_t}, c_i) = c_i$, $\mathbf{\Delta_i} = [-1\ 1]^T$ and $P(\mathbf{x}_t - \mathbf{x}_{t-1} = \mathbf{\Delta_i}) = P(\mathbf{x}_t|\mathbf{x}_{t-1}, v_i) = c_i$.

In a traditional hidden Markov model, the transition kernel is typically fixed. In comparison, SKM is better at capturing dynamic interactions in terms of the events with rates dependent on reactant populations, as shown in Eq.(2).

# 3 Variational Inference with the Stochastic Kinetic Model

In this section, we define the likelihood of the entire sequence of hidden states and observations for an event-based model, and derive a variational inference algorithm and parameter-learning algorithm.

## 3.1 Likelihood for Event-based Model

In social dynamics, we use a discrete time Markov model to describe the temporal evolutions of a set of individuals $x^{(1)}, \ldots, x^{(M)}$ according to a set of $V$ events. To cope with dynamic interactions, we introduce the SKM and express the state transition probabilities in terms of event probabilities, as shown in Figure 2.1(b). We assume at each discrete time step that no more than one event will occur. Let $v_1, \ldots, v_T$ be a sequence of events, $\mathbf{x_1}, \ldots, \mathbf{x_T}$ a sequence of hidden states, and $\mathbf{y_1}, \ldots, \mathbf{y_T}$ a set of observations. Similar to Eq.(1), the likelihood of the entire sequence is as follows:

$$P\left(\mathbf{x}_{1,\ldots,T}, \mathbf{y}_{1,\ldots,T}, v_{1,\ldots,T}\right) = \prod_{t=1}^{T} P(\mathbf{x}_t, v_t | \mathbf{x}_{t-1}) P(\mathbf{y}_t | \mathbf{x}_t), \text{ where} \tag{3}$$

$$P(\mathbf{x}_t, v_t | \mathbf{x}_{t-1}) = \begin{cases} c_k \cdot g_k\left(\mathbf{x}_{t-1}\right) \cdot \delta(\mathbf{x}_t - \mathbf{x}_{t-1} \equiv \mathbf{\Delta_k}) & \text{if } v_t = k \\ (1 - \sum_k c_k g_k\left(\mathbf{x}_{t-1}\right)) \cdot \delta(\mathbf{x}_t - \mathbf{x}_{t-1} \equiv \mathbf{0}) & \text{if } v_t = \emptyset \end{cases}.$$

$P(\mathbf{x}_t, v_t | \mathbf{x}_{t-1})$ is the event-based transition kernel. $\delta(\mathbf{x}_t - \mathbf{x}_{t-1} \equiv \mathbf{\Delta_k})$ is 1 if the previous state is $\mathbf{x}_{t-1}$ and the current state is $\mathbf{x}_t = \mathbf{x}_{t-1} + \mathbf{\Delta_k}$, and 0 otherwise. $\mathbf{\Delta_k}$ is the effect of event $v_k$. $\emptyset$ represents an auxiliary event, meaning that there is no event. Substituting the product form of $g_k$, the transition kernel can be written as follows:

$$P(\mathbf{x}_t, v_t = k | \mathbf{x}_{t-1}) = c_k \prod_m g_k^{(m)}(x_{t-1}^{(m)}) \cdot \prod_m \delta(x_t^{(m)} - x_{t-1}^{(m)} \equiv \Delta_k^{(m)}), \tag{4}$$

$$P(\mathbf{x}_t, v_t = \emptyset | \mathbf{x}_{t-1}) = (1 - \sum_k c_k \prod_m g_k^{(m)}(x_{t-1}^{(m)})) \cdot \prod_m \delta(x_t^{(m)} - x_{t-1}^{(m)} \equiv 0), \tag{5}$$

where $\delta(x_t^{(m)} - x_{t-1}^{(m)} \equiv \Delta_k^{(m)})$ is 1 if the previous state of an individual $m$ is $x_{t-1}^{(m)}$ and the current state is $x_t^{(m)} = x_{t-1}^{(m)} + \Delta_k^{(m)}$, and 0 otherwise.

## 3.2 Variational Inference for Stochastic Kinetic Model

As noted in Section 2.1, exact inference in social dynamics is intractable due to the formidable state space. However, we can approximate the posterior distribution $P(\mathbf{x}_{1,\ldots,T}, v_{1,\ldots,T} | \mathbf{y}_{1,\ldots,T})$ using an approximate distribution within the exponential family. The inference algorithm minimizes the KL divergence between these two distributions, which can be formulated as an optimization problem [14]:

$$\text{Minimize:} \quad \sum_{t, \mathbf{x}_{t-1}, \mathbf{x}_t, v_t} \hat{\xi}_t(\mathbf{x}_{t-1}, \mathbf{x}_t, v_t) \cdot \log \frac{\hat{\xi}_t(\mathbf{x}_{t-1}, \mathbf{x}_t, v_t)}{P(\mathbf{x}_t, v_t | \mathbf{x}_{t-1}) P(\mathbf{y}_t | \mathbf{x}_t)} \tag{6}$$

$$- \sum_{t, \mathbf{x}_t} \prod_m \hat{\gamma}_t^{(m)}(x_t^{(m)}) \log \prod_m \hat{\gamma}_t^{(m)}(x_t^{(m)})$$

$$\text{Subject to:} \quad \sum_{v_t, \mathbf{x}_{t-1}, \{\mathbf{x}_t \backslash x_t^{(m)}\}} \hat{\xi}_t(\mathbf{x}_{t-1}, \mathbf{x}_t, v_t) = \hat{\gamma}_t^{(m)}(x_t^{(m)}), \text{ for all } t, m, x_t^{(m)},$$

$$\sum_{v_t, \{\mathbf{x}_{t-1} \backslash x_{t-1}^{(m)}\}, \mathbf{x}_t} \hat{\xi}_t(\mathbf{x}_{t-1}, \mathbf{x}_t, v_t) = \hat{\gamma}_{t-1}^{(m)}(x_{t-1}^{(m)}), \text{ for all } t, m, x_{t-1}^{(m)},$$

$$\sum_{x_t^{(m)}} \hat{\gamma}_t^{(m)}(x_t^{(m)}) = 1, \text{ for all } t, m.$$

The objective function is the Bethe free energy, composed of average energy and Bethe entropy approximation [23]. $\hat{\xi}_t(\mathbf{x}_{t-1}, \mathbf{x}_t, v_t)$ is the approximate two-slice statistics and $\hat{\gamma}_t^{(m)}(x_t^{(m)})$ is the approximate one-slice statistics for each individual $m$. They form the approximate distribution over which to minimize the Bethe free energy. The $\sum_{t, \mathbf{x}_{t-1}, \mathbf{x}_t, v_t}$ is an abbreviation for summing over $t$, $\mathbf{x}_{t-1}$, $\mathbf{x}_t$, and $v_t$. $\sum_{\{\mathbf{x}_t \backslash x_t^{(m)}\}}$ is the sum over all individuals in $\mathbf{x_t}$ except $x_t^{(m)}$. We use similar abbreviations below. The first two sets of constraints are marginalization conditions, and the third

is normalization conditions. To solve this constrained optimization problem, we first define the Lagrange function using Lagrange multipliers to weight constraints, then take the partial derivatives with respect to $\hat{\xi}_t(\mathbf{x}_{t-1}, \mathbf{x}_t, v_t)$, and $\hat{\gamma}_t^{(m)}(x_t^{(m)})$. The dual problem is to find the approximate forward statistics $\hat{\alpha}_{t-1}^{(m)}(x_{t-1}^{(m)})$ and backward statistics $\hat{\beta}_t^{(m)}(x_t^{(m)})$ in order to maximize the pseudo-likelihood function. The duality is between minimizing Bethe free energy and maximizing pseudo-likelihood. The fixed-point solution for the primal problem is as follows[1]:

$$\hat{\xi}_t(x_{t-1}^{(m)}, x_t^{(m)}, v_t) = \frac{1}{Z_t} \sum_{m' \neq m, x_{t-1}^{(m')}, x_t^{(m')}} P(\mathbf{x}_t, v_t | \mathbf{x}_{t-1}) \cdot \prod_m \hat{\alpha}_{t-1}^{(m)}(x_{t-1}^{(m)}) \cdot \prod_m P(y_t^{(m)} | x_t^{(m)}) \cdot \prod_m \hat{\beta}_t^{(m)}(x_t^{(m)}). \tag{7}$$

$\hat{\xi}_t(x_{t-1}^{(m)}, x_t^{(m)}, v_t)$ is the two-slice statistics for an individual $m$, and $Z_t$ is the normalization constant. Given the factorized form of $P(\mathbf{x}_t, v_t | \mathbf{x}_{t-1})$ in Eqs. (4) and (5), everything in Eq. (7) can be written in a factorized form. After reformulating the term relevant to the individual $m$, $\hat{\xi}_t(x_{t-1}^{(m)}, x_t^{(m)}, v_t)$ can be shown neatly as follows:

$$\hat{\xi}_t(x_{t-1}^{(m)}, x_t^{(m)}, v_t) = \frac{1}{Z_t} \hat{P}(x_t^{(m)}, v_t | x_{t-1}^{(m)}) \cdot \hat{\alpha}_{t-1}^{(m)}(x_{t-1}^{(m)}) P(y_t^{(m)} | x_t^{(m)}) \hat{\beta}_t^{(m)}(x_t^{(m)}), \tag{8}$$

where the marginalized transition kernel $\hat{P}(x_t^{(m)}, v_t | x_{t-1}^{(m)})$ for the individual $m$ can be defined as:

$$\hat{P}(x_t^{(m)}, v_t = k | x_{t-1}^{(m)}) = c_k g_k^{(m)}(x_{t-1}^{(m)}) \prod_{m' \neq m} \tilde{g}_{k,t-1}^{(m')} \cdot \delta(x_t^{(m)} - x_{t-1}^{(m)} \equiv \Delta_k^{(m)}), \tag{9}$$

$$\hat{P}(x_t^{(m)}, v_t = \emptyset | x_{t-1}^{(m)}) = (1 - \sum_k c_k g_k^{(m)}(x_{t-1}^{(m)}) \prod_{m' \neq m} \hat{g}_{k,t-1}^{(m')}) \delta(x_t^{(m)} - x_{t-1}^{(m)} \equiv 0), \tag{10}$$

$$\tilde{g}_{k,t-1}^{(m')} = \sum_{x_t^{(m')} - x_{t-1}^{(m')} \equiv \Delta_k^{(m')}} \alpha_{t-1}^{(m')}(x_{t-1}^{(m')}) P(y_t^{(m')} | x_t^{(m')}) \beta_t^{(m')}(x_t^{(m')}) g_k^{(m')}(x_{t-1}^{(m')}) \Big/ \sum_{x_t^{(m')} - x_{t-1}^{(m')} \equiv 0} \alpha_{t-1}^{(m')}(x_{t-1}^{(m')}) P(y_t^{(m')} | x_t^{(m')}) \beta_t^{(m')}(x_t^{(m')}),$$

$$\hat{g}_{k,t-1}^{(m')} = \sum_{x_t^{(m')} - x_{t-1}^{(m')} \equiv 0} \alpha_{t-1}(x_{t-1}^{(m')}) P(y_t^{(m')} | x_t^{(m')}) \beta_t^{(m')}(x_t^{(m')}) g_k^{(m')}(x_{t-1}^{(m')}) \Big/ \sum_{x_t^{(m')} - x_{t-1}^{(m')} \equiv 0} \alpha_{t-1}^{(m')}(x_{t-1}^{(m')}) P(y_t^{(m')} | x_t^{(m')}) \beta_t^{(m')}(x_t^{(m')}),$$

In the above equations, we consider the mean field effect by summing over the current and previous states of all the other individuals $m' \neq m$. The marginalized transition kernel considers the probability of event $k$ on the individual $m$ given the context of the temporal evolutions of the other individuals. Comparing Eqs. (9) and (10) with Eqs. (4) and (5), instead of multiplying $g_k^{(m')}(x_{t-1}^{(m')})$ for individual $m' \neq m$, we use the expected value of $g_k^{(m')}$ with respect to the marginal probability distribution of $x_{t-1}^{(m')}$.

**Complexity Analysis**: In our inference algorithm, the most computation-intensive step is the marginalization in Eqs. (9)-(10). The complexity is $O(MS^2)$, where $M$ is the number of individuals and $S$ is the state space of a single individual. The complexity of the entire algorithm is therefore $O(MS^2TN)$, where $T$ is the number of time steps and $N$ is the number of iterations until convergence. As such, the complexity of our algorithm grows only linearly with the number of individuals; it offers excellent scalability when the number of tracked individuals becomes large.

### 3.3 Parameter Learning

In order to learn the rate constant $c_k$, we maximize the expected log likelihood. In a stochastic kinetic model, the probability of a sample path is given in Eq. (3). The expected log likelihood over the posterior probability conditioned on the observations $\mathbf{y}_1, \ldots, \mathbf{y}_T$ takes the following form:

$$\log P(\mathbf{x}_{1,\ldots,T}, \mathbf{y}_{1,\ldots,T}, v_{1,\ldots,T}) = \sum_{t, \mathbf{x}_{t-1}, \mathbf{x}_t, v_t} \hat{\xi}_t(\mathbf{x}_{t-1}, \mathbf{x}_t, v_t) \cdot \log(P(\mathbf{x}_t, v_t | \mathbf{x}_{t-1}) P(\mathbf{y}_t | \mathbf{x}_t)).$$

$\hat{\xi}_t(\mathbf{x}_{t-1}, \mathbf{x}_t, v_t)$ is the approximate two-slice statistics defined in Eq. (6). Maximizing this expected log likelihood by setting its partial derivative over the rate constants to 0 gives the maximum expected log likelihood estimation of these rate constants.

$$c_k = \frac{\sum_{t, \mathbf{x}_{t-1}, \mathbf{x}_t} \hat{\xi}_t(\mathbf{x}_{t-1}, \mathbf{x}_t, v_t = k)}{\sum_{t, \mathbf{x}_{t-1}, \mathbf{x}_t} \hat{\xi}_t(\mathbf{x}_{t-1}, \mathbf{x}_t, v_t = \emptyset) g_k(\mathbf{x}_{t-1})} \approx \frac{\sum_t \sum_{\mathbf{x}_{t-1}, \mathbf{x}_t} \hat{\xi}_t(\mathbf{x}_{t-1}, \mathbf{x}_t, v_t = k)}{\sum_t \prod_m \sum_{x_{t-1}^{(m)}} \hat{\gamma}_{t-1}^{(m)}(x_{t-1}^{(m)}) g_k^{(m)}(x_{t-1}^{(m)})}. \tag{11}$$

As such, the rate constant for event $k$ is the expected number of times that this event has occurred divided by the total expected number of times this event could have occurred.

To summarize, we provide the variational inference algorithm below.

---

**Algorithm: Variational Inference with a Stochastic Kinetic Model**

Given the observations $y_t^{(m)}$ for $t = 1, \ldots, T$ and $m = 1, \ldots, M$, find $x_t^{(m)}$, $v_t$ and rate constants $c_k$ for $k = 1, \ldots, V$.

**Latent state inference.** Iterate through the following forward and backward passes until convergence, where $\hat{P}(x_t^{(m)}, v_t | x_{t-1}^{(m)})$ is given by Eqs. (9) and (10).

- Forward pass. For $t = 1, \ldots, T$ and $m = 1, \ldots, M$, update $\hat{\alpha}_t^{(m)}(x_t^{(m)})$ according to

$$\hat{\alpha}_t^{(m)}(x_t^{(m)}) \leftarrow \frac{1}{Z_t} \sum_{x_{t-1}^{(m)}, v_t} \hat{\alpha}_{t-1}^{(m)}(x_{t-1}^{(m)}) \hat{P}(x_t^{(m)}, v_t | x_{t-1}^{(m)}) P(y_t^{(m)} | x_t^{(m)}).$$

- Backward pass. For $t = T, \ldots, 1$ and $m = 1, \ldots, M$, update $\hat{\beta}_{t-1}^{(m)}(x_{t-1}^{(m)})$ according to

$$\hat{\beta}_{t-1}^{(m)}(x_{t-1}^{(m)}) \leftarrow \frac{1}{Z_t} \sum_{x_t^{(m)}, v_t} \hat{\beta}_t^{(m)}(x_t^{(m)}) \hat{P}(x_t^{(m)}, v_t | x_{t-1}^{(m)}) P(y_t^{(m)} | x_t^{(m)}).$$

**Parameter estimation.** Iterate through the latent state inference (above) and rate constants estimate of $c_k$ according to Eq. (11), until convergence.

---

## 4   Experiments on Epidemic Applications

In this section, we evaluate the performance of variational inference with a stochastic kinetic model (VISKM) algorithm of epidemic dynamics, with which we predict the transmission of diseases and the health status of each individual based on proximity data collected from sensor networks.

### 4.1   Epidemic Dynamics

In epidemic dynamics, $G_t = (\mathcal{M}, E_t)$ is a dynamic network, where each node $m \in \mathcal{M}$ is an individual in the network, and $E_t = \{(m_i, m_j)\}$ is a set of edges in $G_t$ representing that individuals $m_i$ and $m_j$ have interacted at a specific time $t$. There are two possible hidden states for each individual $m$ at time $t$, $x_t^{(m)} \in \{0, 1\}$, where 0 indicates the susceptible state and 1 the infectious state. $y_t^{(m)} \in \{0, 1\}$ represents the presence or absence of symptoms for individual $m$ at time $t$. $P(y_t^{(m)} | x_t^{(m)})$ represents the observation probability. We define three types of events in epidemic applications: (1) A previously infectious individual recovers and becomes susceptible again: $I \xrightarrow{c_1} S$. (2) An infectious individual infects a susceptible individual in the network: $S + I \xrightarrow{c_2} 2I$. (3) A susceptible individual in the network is infected by an outside infectious individual: $S \xrightarrow{c_3} I$. Based on these events, the transition kernel can be defined as follows:

$$P(x_t^{(m)} = 0 | x_{t-1}^{(m)} = 1) = c_1, \ P(x_t^{(m)} = 1 | x_{t-1}^{(m)} = 1) = 1 - c_1,$$

$$P(x_t^{(m)} = 0 | x_{t-1}^{(m)} = 0) = (1 - c_3)(1 - c_2)^{C_{m,t}}, \ P(x_t^{(m)} = 1 | x_{t-1}^{(m)} = 0) = 1 - (1 - c_3)(1 - c_2)^{C_{m,t}},$$

where $C_{m,t} = \sum_{m':(m',m) \in E_t} \delta(x_t^{(m')} \equiv 1)$ is the number of possible infectious sources for individual $m$ at time $t$. Intuitively, the probability of a susceptible individual becoming infected is 1 minus the probability that no infectious individuals (inside or outside the network) infected him. When the probability of infection is very small, we can approximate $P(x_t^{(m)} = 1 | x_{t-1}^{(m)} = 0) \approx c_3 + c_2 \cdot C_{m,t}$.

## 4.2 Experimental Results

**Data Explanation:** We employ two data sets of epidemic dynamics. The real data set is collected from the Social Evolution experiment [5, 6]. This study records "common cold" symptoms of 65 students living in a university residence hall from January 2009 to April 2009, tracking their locations and proximities using mobile phones. In addition, the students took periodic surveys regarding their health status and personal interactions. The synthetic data set was collected on the Dartmouth College campus from April 2001 to June 2004, and contains the movement history of 13,888 individuals [16]. We synthesized disease transmission along a timeline using the popular susceptible-infectious-susceptible (SIS) epidemiology model [15], then applied the VISKM to calibrate performance. We selected this data set because we want to demonstrate that our model works on data with a large number of people over a long period of time.

**Evaluation Metrics and Baseline Algorithms:** We select the receiver operating characteristic (ROC) curve as our performance metric because the discrimination thresholds of diseases vary. We first compare the accuracy and efficiency of VISKM with Gibbs sampling (Gibbs) and particle filtering (PF) on the Social Evolution data set [7, 8].[2] Both Gibbs sampling and particle filtering iteratively sample the infectious and susceptible latent state sequences and the infection and recovery events conditioned on these state sequences. Gibbs-Prediction-10000 indicates 10,000 iterations of Gibbs sampling with 1000 burn-in iterations for the prediction task. PF-Smoothing-1000 similarly refers to 1000 iterations of particle filtering for the smoothing task. All experiments are performed on the same computer.

**Individual State Inference:** We infer the probabilities of a hidden infectious state for each individual at different times under different scenarios. There are three tasks: 1. *Prediction*: Given an individual's past health and current interaction patterns, we predict the current infectious latent state. Figure 2(a) compares prediction performance among the different approximate inference methods. 2. *Smoothing:* Given an individual's interaction patterns and past health with missing periods, we infer the infectious latent states during these missing periods. Figure 2(b) compares the performance of the three inference methods. 3. *Expansion*: Given the health records of a portion ($\sim 10\%$) of the population, we estimate the individual infectious states of the entire population before medically inspecting them. For example, given either a group of volunteers willing to report their symptoms or the symptom data of patients who came to hospitals, we determine the probabilities that the people near these individuals also became or will become infected. This information helps the government or aid agencies to efficiently distribute limited medical resources to those most in need. Figure 2(c) compares the performance of the different methods. From the above three graphs, we can see that all three methods identify the infectious states in an accurate way. However, VISKM outperforms Gibbs sampling and particle filtering in terms of area under the ROC curve for all three tasks. VISKM has an advantage in the smoothing task because the backward pass helps to infer the missing states using subsequent observations. In addition, the performance of Gibbs and PF improves as the number of samples/particles increases.

Figure 2(d) shows the performance of the three tasks on the Dartmouth data set. We do not apply the same comparison because it takes too much time for sampling. From the graph, we can see that VISKM infers most of the infectious moments of individuals in an accurate way for a large social system. In addition, the smoothing results are slightly better than the prediction results because we can leverage observations from both directions. The expansion case is *relatively* poor, because we use only very limited information to derive the results; however, even in this case the ROC curve has good discriminating power to differentiate between infectious and susceptible individuals.

**Collective Statistics Inference:** After determining the individual results, we aggregate them to approximate the total number of infected individuals in the social system as time evolves. This offers a collective statistical summary of the spread of disease in one area as in traditional research, which typically scales the sample statistics with respect to the sample ratio. Figures 2(e) and (f) show that given $20\%$ of the Social Evolution data and $10\%$ of the Dartmouth data, VISKM estimates the collective statistics better than the other methods.

**Efficiency and Scalability:** Table 1 shows the running time of different algorithms for the Social Evolution data on the same computer. From the table, we can see that Gibbs sampling runs slightly longer than PF, but they are in the same scale. However, VISKM requires much less computation time.

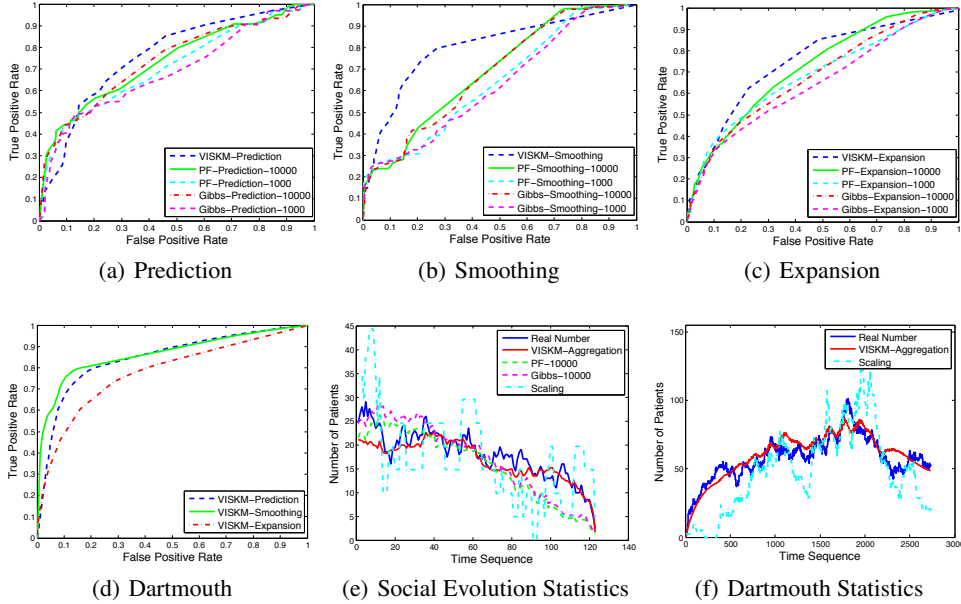

Figure 2: Experimental results. (a-c) show the prediction, smoothing, and expansion performance comparisons for Social Evolution data, while (d) shows performance of the three tasks for Dartmouth data. (e-f) represent the statistical inferences for both data sets.

Table 1: Running time for different approximate inference algorithms. Gibbs_10000 refers to Gibbs sampling for 10,000 iterations, and PF_1000 to particle filtering for 1000 iterations. Other entries follow the same pattern. All times are measured in seconds.

|            | VISKM | Gibbs_1000 | Gibbs_10000 | PF_1000 | PF_10000 |
|------------|-------|------------|-------------|---------|----------|
| 60 People  | 0.78  | 771        | 7820        | 601     | 6100     |
| 30 People  | 0.39  | 255        | 2556        | 166     | 1888     |
| 15 People  | 0.19  | 101        | 1003        | 122     | 1435     |

In addition, the computation time of VISKM grows linearly with the number of individuals, which validates the complexity analysis in Section 3.2. Thus, it offers excellent scalability for large social systems. In comparison, Gibbs sampling and PF grow super linearly with the number of individuals, and roughly linearly with the number of samples.

**Summary:** Our proposed VISKM achieves higher accuracy in terms of area under ROC curve and collective statistics than Gibbs sampling or particle filtering (within 10,000 iterations). More importantly, VISKM is more efficient than sampling with much less computation time. Additionally, the computation time of VISKM grows linearly with the number of individuals, demonstrating its excellent scalability for large social systems.

## 5   Conclusions

In this paper, we leverage sensor network and social network data to capture temporal evolution in social dynamics and infer individual behaviors. In order to define the adaptive transition kernel, we introduce a stochastic dynamic mode that captures the dynamics of complex interactions. In addition, in order to make tractable inferences we propose a variational inference algorithm the computation complexity of which grows linearly with the number of individuals. Large-scale experiments on epidemic dynamics demonstrate that our method effectively captures the evolution of social dynamics and accurately infers individual behaviors. More accurate collective effects can be also derived through the aggregated results. Potential applications for our algorithm include the dynamics of emotion, opinion, rumor, collaboration, and friendship.

## Footnotes

[1]The derivations for the optimization problem and its solution are shown in the Supplemental Material.

[2]Code and data are available at `http://cse.buffalo.edu/~wendong/`.

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
