[Supplementary Material · supplement.pdf]

# 6 Appendix

## 6.1 Derivation of the optimization problem in Eq.(6)

Let $P(\mathbf{x}_{1,...,T}, v_{1,...,T}|\mathbf{y}_{1,...,T})$ be the exact posterior. Our goal is to approximate this posterior by a distribution $Q(\mathbf{x}_{1,...,T}, v_{1,...,T})$ in the exponential family that minimizes the KL divergence between these two distributions:

$$KL(Q(\mathbf{x}_{1,...,T}, v_{1,...,T})|P(\mathbf{x}_{1,...,T}, v_{1,...,T}|\mathbf{y}_{1,...,T}))$$

$$= \sum_{\mathbf{x}_{1,...,T}, v_{1,...,T}} Q(\mathbf{x}_{1,...,T}, v_{1,...,T}) \log[\frac{Q(\mathbf{x}_{1,...,T}, v_{1,...,T}) \cdot P(\mathbf{y}_{1,...,T})}{P(\mathbf{x}_{1,...,T}, \mathbf{y}_{1,...,T}, v_{1,...,T})}]$$

$$= \sum_{\mathbf{x}_{1,...,T}, v_{1,...,T}} Q(\mathbf{x}_{1,...,T}, v_{1,...,T}) \log Q(\mathbf{x}_{1,...,T}, v_{1,...,T})$$

$$- \sum_{t=1}^{T} \sum_{\mathbf{x}_{1,...,T}, v_{1,...,T}} Q(\mathbf{x}_{1,...,T}, v_{1,...,T}) \log P(\mathbf{x}_t, \mathbf{y}_t, v_t|\mathbf{x}_{t-1}). \tag{12}$$

In the first step, we apply the definition of conditional probability and KL-divergence. In the second, we omit $P(\mathbf{y}_{1,...,T})$ because it is a constant in this optimization problem. In addition, we decompose $P(\mathbf{x}_{1,...,T}, \mathbf{y}_{1,...,T}, v_{1,...,T}) = \prod_{t=1}^{T} P(\mathbf{x}_t, \mathbf{y}_t, v_t|\mathbf{x}_{t-1})$.

We then define the approximate two-slice statistics $\hat{\xi}_t(\mathbf{x}_{t-1}, \mathbf{x}_t, v_t)$ and one-slice statistics $\hat{\gamma}_t(\mathbf{x}_t)$. Both are in the exponential family. In this context, we have $M$ individuals in the system and the mean-field approximation can be shown as $\hat{\gamma}_t(\mathbf{x}_t) = \prod_{m=1}^{N} \hat{\gamma}_t^{(m)}(x_t^{(m)})$, where $\hat{\gamma}_t^{(m)}(x_t^{(m)})$ is the approximate one-slice statistics for individual $m$. Given the observation that $Q(\mathbf{x}_{1,...,T}, v_{1,...,T})$ can be expressed as a product of two-slice statistics divided by a product of one-slice statistics, then

$$Q(\mathbf{x}_{1,...,T}, v_{1,...,T}) = \frac{\prod_{t=1}^{T} \hat{\xi}_t(\mathbf{x}_{t-1}, \mathbf{x}_t, v_t)}{\prod_{t=1}^{T-1} \hat{\gamma}_t(\mathbf{x}_t)} = \frac{\prod_{t=1}^{T} \hat{\xi}_t(\mathbf{x}_{t-1}, \mathbf{x}_t, v_t)}{\prod_{t=1}^{T-1} \prod_{m=1}^{M} \hat{\gamma}_t^{(m)}(x_t^{(m)})}. \tag{13}$$

If we substitute Eq. (13) into Eq. (12), the objective function becomes the following:

$$\sum_{\mathbf{x}_{1,...,T}, v_{1,...,T}} Q(\mathbf{x}_{1,...,T}, v_{1,...,T}) \log \frac{\prod_{t=1}^{T} \hat{\xi}_t(\mathbf{x}_{t-1}, \mathbf{x}_t, v_t)}{\prod_{t=1}^{T-1} \prod_m \hat{\gamma}_t^{(m)}(x_t^{(m)})}$$

$$- \sum_{t=1}^{T} \sum_{\mathbf{x}_{1,...,T}, v_{1,...,T}} Q(\mathbf{x}_{1,...,T}, v_{1,...,T}) \log P(\mathbf{x}_t, \mathbf{y}_t, v_t|\mathbf{x}_{t-1})$$

$$= \sum_{t,\mathbf{x}_{t-1}, \mathbf{x}_t, v_t} \hat{\xi}_t(\mathbf{x}_{t-1}, \mathbf{x}_t, v_t) \log \frac{\hat{\xi}_t(\mathbf{x}_{t-1}, \mathbf{x}_t, v_t)}{P(\mathbf{x}_t, \mathbf{y}_t, v_t|\mathbf{x}_{t-1})}$$

$$- \sum_{t,\mathbf{x}_t} \prod_m \hat{\gamma}_t^{(m)}(x_t^{(m)}) \log \prod_m \hat{\gamma}_t^{(m)}(x_t^{(m)}). \tag{14}$$

This objective function is subject to marginalization and normalization constraints:

$$\sum_{v_t, \mathbf{x}_{t-1}, \{\mathbf{x}_t \backslash x_t^{(m)}\}} \hat{\xi}_t(\mathbf{x}_{t-1}, \mathbf{x}_t, v_t) = \hat{\gamma}_t^{(m)}(x_t^{(m)}), \text{ for all } t, m, x_t^{(m)},$$

$$\sum_{v_t, \{\mathbf{x}_{t-1} \backslash x_{t-1}^{(m)}\}, \mathbf{x}_t} \hat{\xi}_t(\mathbf{x}_{t-1}, \mathbf{x}_t, v_t) = \hat{\gamma}_{t-1}^{(m)}(x_{t-1}^{(m)}), \text{ for all } t, m, x_{t-1}^{(m)},$$

$$\sum_{x_t^{(m)}} \hat{\gamma}_t^{(m)}(x_t^{(m)}) = 1, \text{ for all } t, m.$$

$\sum_{\{\mathbf{x}_t \backslash x_t^{(m)}\}}$ refers to the sum over all values of $\mathbf{x}_t$ except $x_t^{(m)}$.

## 6.2 Derivation of the inference algorithm from Eq.(8) to Eq.(10)

The optimization problem derived from Eq. (14) along with the constraints can be shown as follows:

$$\sum_{t,\mathbf{x}_{t-1},\mathbf{x}_t,v_t} \hat{\xi}_t(\mathbf{x}_{t-1},\mathbf{x}_t,v_t) \log \frac{\hat{\xi}_t(\mathbf{x}_{t-1},\mathbf{x}_t,v_t)}{P(\mathbf{x}_t,\mathbf{y}_t,v_t|\mathbf{x}_{t-1})} - \sum_{t,\mathbf{x}_t} \prod_m \hat{\gamma}_t^{(m)}(x_t^{(m)}) \log \prod_m \hat{\gamma}_t^{(m)}(x_t^{(m)}) \quad (15)$$

subject to:

$$\sum_{v_t,\mathbf{x}_{t-1},\{\mathbf{x}_t \setminus x_t^{(m)}\}} \hat{\xi}_t(\mathbf{x}_{t-1},\mathbf{x}_t,v_t) = \hat{\gamma}_t^{(m)}(x_t^{(m)}), \text{ for all } t, m, x_t^{(m)},$$

$$\sum_{v_t,\{\mathbf{x}_{t-1} \setminus x_{t-1}^{(m)}\},\mathbf{x}_t} \hat{\xi}_t(\mathbf{x}_{t-1},\mathbf{x}_t,v_t) = \hat{\gamma}_{t-1}^{(m)}(x_{t-1}^{(m)}), \text{ for all } t, m, x_{t-1}^{(m)},$$

$$\sum_{x_t^{(m)}} \hat{\gamma}_t^{(m)}(x_t^{(m)}) = 1, \text{ for all } t, m.$$

We apply the method of Lagrange multipliers to solve this, which begins with forming the Lagrange function to be optimized:

$$L = \sum_{t,\mathbf{x}_{t-1},\mathbf{x}_t,v_t} \hat{\xi}_t(\mathbf{x}_{t-1},\mathbf{x}_t,v_t) \log \frac{\hat{\xi}_t(\mathbf{x}_{t-1},\mathbf{x}_t,v_t)}{P(\mathbf{x}_t,\mathbf{y}_t,v_t|\mathbf{x}_{t-1})} - \sum_{t,\mathbf{x}_t} \prod_m \hat{\gamma}_t^{(m)}(x_t^{(m)}) \log \prod_m \hat{\gamma}_t^{(m)}(x_t^{(m)}) \quad (16)$$

$$+ \sum_{t,m,x_t^{(m)}} \lambda_t^{(m)}(x_t^{(m)}) \left( \sum_{v_t,\mathbf{x}_{t-1},\{\mathbf{x}_t \setminus x_t^{(m)}\}} \hat{\gamma}_t^{(m)}(x_t^{(m)}) - \hat{\xi}_t(\mathbf{x}_{t-1},\mathbf{x}_t,v_t) \right)$$

$$+ \sum_{t,m,x_{t-1}^{(m)}} \mu_{t-1}^{(m)}(x_{t-1}^{(m)}) \left( \sum_{v_t,\{\mathbf{x}_{t-1} \setminus x_{t-1}^{(m)}\},\mathbf{x}_t} \hat{\gamma}_{t-1}^{(m)}(x_{t-1}^{(m)}) - \hat{\xi}_t(\mathbf{x}_{t-1},\mathbf{x}_t,v_t) \right) .$$

$$+ \sum_{t,m,x_t^{(m)}} \nu(x_t^{(m)}) \left( \sum_{x_t^{(m)}} \hat{\gamma}_t^{(m)}(x_t^{(m)}) - 1 \right)$$

We then set the partial derivatives of Eq. (16) over $\hat{\xi}_t(\mathbf{x}_{t-1},\mathbf{x}_t,v_t)$ to 0, which results in the following:

$$\frac{\partial L}{\partial \hat{\xi}_t(\mathbf{x}_{t-1},\mathbf{x}_t,v_t)} = \log \frac{\hat{\xi}_t(\mathbf{x}_{t-1},\mathbf{x}_t,v_t)}{P(\mathbf{x}_t,\mathbf{y}_t,v_t|\mathbf{x}_{t-1})} + 1 - \sum_m \lambda_t^{(m)}(x_t^{(m)}) - \sum_m \mu_{t-1}^{(m)}(x_{t-1}^{(m)}) \stackrel{\text{set}}{=} 0$$

$$\Rightarrow \hat{\xi}_t(\mathbf{x}_{t-1},\mathbf{x}_t,v_t) \propto \exp \left( \sum_m \mu_{t-1}^{(m)}(x_{t-1}^{(m)}) \right) P(\mathbf{x}_t,\mathbf{y}_t,v_t|\mathbf{x}_{t-1}) \exp \left( \sum_m \lambda_t^{(m)}(x_t^{(m)}) \right),$$

As such, we see that $\hat{\alpha}_{t-1}^{(m)}(x_{t-1}^{(m)}) = \exp(\mu_{t-1}^{(m)}(x_{t-1}^{(m)}))$ is associated with the forward probabilities and $\hat{\beta}_t^{(m)}(x_t^{(m)}) = \exp(\lambda_t^{(m)}(x_t^{(m)}))$ with the backward probabilities, with $\hat{\gamma}_t^{(m)}(x_t^{(m)}) = \hat{\alpha}_t^{(m)}(x_t^{(m)})\hat{\beta}_t^{(m)}(x_t^{(m)})$. We can determine the two-slice statistics for an individual $m$ by marginalizing the other individuals $m' \neq m$:

$$\hat{\xi}_t(x_{t-1}^{(m)},x_t^{(m)},v_t) = \sum_{m' \neq m,x_{t-1}^{(m')},x_t^{(m')}} \hat{\xi}_t(\mathbf{x}_{t-1},\mathbf{x}_t,v_t)$$

$$\propto \sum_{m' \neq m,x_{t-1}^{(m')},x_t^{(m')}} P(\mathbf{x}_t,v_t|\mathbf{x}_{t-1}) \cdot \prod_m \hat{\alpha}_{t-1}^{(m)}(x_{t-1}^{(m)}) \cdot \prod_m P(y_t^{(m)}|x_t^{(m)}) \cdot \prod_m \hat{\beta}_t^{(m)}(x_t^{(m)}).$$

The above is the same as in Eq. (7).

## 6.3 Derivation of the parameter-learning algorithm

From Eq.(3), the log-likelihood of the entire sequence can be shown as this:

$$\log P\left(\mathbf{x}_{1,\dots,T}, \mathbf{y}_{1,\dots,T}, v_{1,\dots,T}\right) = \sum_{t=1}^{T} \log P(\mathbf{x}_t, v_t | \mathbf{x}_{t-1}) + \sum_{t=1}^{T} \log P(\mathbf{y}_t | \mathbf{x}_t), \text{ where} \quad (17)$$

$$P(\mathbf{x}_t, v_t | \mathbf{x}_{t-1}) = \begin{cases} c_k \cdot g_k(\mathbf{x}_{t-1}) \cdot \delta(\mathbf{x}_t - \mathbf{x}_{t-1} \equiv \mathbf{\Delta_k}) & \text{if } v_t = k \\ (1 - \sum_k c_k g_k(\mathbf{x}_{t-1})) \cdot \delta(\mathbf{x}_t - \mathbf{x}_{t-1} \equiv \mathbf{0}) & \text{if } v_t = \emptyset \end{cases}.$$

The probabilities for state transition can be shown as the probabilities of a set of events. The expected log likelihood over the posterior probability conditioned on the observations $\mathbf{y}_1, \dots, \mathbf{y}_T$ takes the following form:

$$\mathbf{E}_{P(\mathbf{x}_{1,\dots,T}, v_{1,\dots,T} | \mathbf{y}_{1,\dots,T})} \left(\log P\left(\mathbf{x}_{1,\dots,T}, \mathbf{y}_{1,\dots,T}, v_{1,\dots,T}\right)\right) \quad (18)$$

$$= \sum_{t,\mathbf{x}_{t-1},\mathbf{x}_t,v_t} \hat{\xi}_t(\mathbf{x}_{t-1}, \mathbf{x}_t, v_t) \cdot \log\left(P(\mathbf{x}_t, v_t | \mathbf{x}_{t-1}) P(\mathbf{y}_t | \mathbf{x}_t)\right)$$

$$= \sum_{t,\mathbf{x}_{t-1},\mathbf{x}_t} \hat{\xi}_t(\mathbf{x}_{t-1}, \mathbf{x}_t, v_t = v) \cdot \log\left(P(\mathbf{x}_t, v_t = v | \mathbf{x}_{t-1}) P(\mathbf{y}_t | \mathbf{x}_t)\right)$$

$$+ \sum_{t,\mathbf{x}_{t-1},\mathbf{x}_t} \hat{\xi}_t(\mathbf{x}_{t-1}, \mathbf{x}_t, v_t = \emptyset) \cdot \log\left(P(\mathbf{x}_t, v_t = \emptyset | \mathbf{x}_{t-1}) P(\mathbf{y}_t | \mathbf{x}_t)\right)$$

At a given time $t$, there are two possible cases: $v_t = v$, where $v \in \{1, \dots, V\}$, and $v_t = \emptyset$. The derivatives with respect to $c_k$ can be shown as follows:

$$\frac{\partial \log P(\mathbf{x}_t, v_t = k | \mathbf{x}_{t-1})}{\partial c_k} = \frac{1}{c_k}$$

$$\frac{\partial \log P(\mathbf{x}_t, v_t = \emptyset | \mathbf{x}_{t-1})}{\partial c_k} = \frac{-g_k(\mathbf{x}_{t-1})}{1 - \sum_k c_k g_k(\mathbf{x}_{t-1})}$$

Note that here we do not detail $\delta(\mathbf{x}_t - \mathbf{x}_{t-1} \equiv \mathbf{\Delta_k})$ and $\delta(\mathbf{x}_t - \mathbf{x}_{t-1} \equiv \mathbf{0})$ explicitly, because when calculating the derivatives of expected log likelihood in Eq.(18) these terms will be contained in $\hat{\xi}_t(\mathbf{x}_{t-1}, \mathbf{x}_t, v_t = k)$ and $\hat{\xi}_t(\mathbf{x}_{t-1}, \mathbf{x}_t, v_t = \emptyset)$. Next we take the derivative of expected log likelihood with respect to $c_k$:

$$\frac{\mathbf{E}_{P(\mathbf{x}_{1,\dots,T}, v_{1,\dots,T} | \mathbf{y}_{1,\dots,T})} \left(\log P\left(\mathbf{x}_{1,\dots,T}, \mathbf{y}_{1,\dots,T}, v_{1,\dots,T}\right)\right)}{\partial c_k} \quad (19)$$

$$= \sum_{t,\mathbf{x}_{t-1},\mathbf{x}_t} \hat{\xi}_t(\mathbf{x}_{t-1}, \mathbf{x}_t, v_t = k) \frac{1}{c_k} - \sum_{t,\mathbf{x}_{t-1},\mathbf{x}_t,} \hat{\xi}_t(\mathbf{x}_{t-1}, \mathbf{x}_t, v_t = \emptyset) \frac{g_k(\mathbf{x}_{t-1})}{1 - \sum_k c_k g_k(\mathbf{x}_{t-1})}$$

Because we assume that the auxiliary event dominates when the time step is small, we approximate $1 - \sum_k c_k g_k(\mathbf{x}_t) \approx 1$ and $\sum_{\mathbf{x}_t} \hat{\xi}_t(\mathbf{x}_{t-1}, \mathbf{x}_t, v_t = \emptyset) \approx \hat{\gamma}_{t-1}(\mathbf{x}_{t-1})$. After applying this approximation and setting the derivative to 0, the result is as follows:

$$c_k = \frac{\sum_t \sum_{\mathbf{x}_{t-1},\mathbf{x}_t} \hat{\xi}_t(\mathbf{x}_{t-1}, \mathbf{x}_t, v_t = k)}{\sum_t \sum_{\mathbf{x}_{t-1},\mathbf{x}_t} \hat{\xi}_t(\mathbf{x}_{t-1}, \mathbf{x}_t, v_t = \emptyset) g_k(\mathbf{x}_{t-1})} \quad (20)$$

$$\approx \frac{\sum_t \sum_{\mathbf{x}_{t-1},\mathbf{x}_t} \hat{\xi}_t(\mathbf{x}_{t-1}, \mathbf{x}_t, v_t = k)}{\sum_t \sum_{\mathbf{x}_{t-1}} \hat{\gamma}_{t-1}(\mathbf{x}_{t-1}) g_k(\mathbf{x}_{t-1})}$$

$$= \frac{\sum_t \sum_{\mathbf{x}_{t-1},\mathbf{x}_t} \hat{\xi}_t(\mathbf{x}_{t-1}, \mathbf{x}_t, v_t = k)}{\sum_t \prod_m \sum_{x_{t-1}^{(m)}} \hat{\gamma}_{t-1}^{(m)}(x_{t-1}^{(m)}) g_k^{(m)}(x_{t-1}^{(m)})}.$$