[Reviews · NeurIPS 2016]

Reviewer 1

Summary

The authors propose that the stochastic kinetic model used in chemistry and cell biology can be used in modeling the spread of things in human societies. They then develop a variational inference algorithm for the model and show its efficacy on mobile phone captured movement data.

Qualitative Assessment

This is a nice paper that goes through the setting up of the SKM model and the deriving the variational bounds. I personally think the paper would have been stronger if it had simply highlighted the "variational inference with stochastic kinetic model" part rather than spend so much space on the social dynamics. It wasn't fully clear, but from what I gathered, no one has developed the variational inference procedure even in the chemistry and biology domains, so the main contribution is on that derivation. With some content reduced on the sociology, the derivation in the supplemental material can be included in the main paper. The framing of the paper could still include some of the social dynamics stuff with the applications mentioned in line 282 and also still include the same experimental results. The mathematics appear to be correct, the experiments sensibly conducted, and the idea good.

Confidence in this Review

2-Confident (read it all; understood it all reasonably well)


Reviewer 2

Summary

This paper focuses on using social dynamics to make prediction on the individual level. To this end, the authors proposed a model that incorporates the stochastic kinetic model into a coupled HMM for characterizing dynamic interactions among the individuals. Further, the paper proposed an efficient variational inference algorithm for making inferences. The proposed model is evaluated on a testbed from epidemic dynamics experiments. The results showed that the proposed model is promising in predicting the disease transmission and the individual infection.

Qualitative Assessment

The major contribution of this paper is three-fold. First, the paper leveraged the stochastic kinetic model to define the transition kernel in the dynamic Bayesian network. Second, the paper proposed a tractable algorithm to make inference on the individual level. Third, the complexity of the inference algorithm grows linearly with the number of individuals. Overall, the proposed model seems reasonable, the algorithm design seems solid, and the experiments seem well executed. However, I feel its novelty in methodologies does not meet the NIPS standard --- the construction of variational inference algorithm is quite standard.

Confidence in this Review

2-Confident (read it all; understood it all reasonably well)


Reviewer 3

Summary

The paper introduces a stochastic kinetic model that is combined with a variational inference algorithm in order to model social dynamics on an individual centre. The authors initially contrast Hidden Markov models and Stochastic Kinetic Models, before formally deriving their model. The essential benefit of the model is that it scales linearly with the number of observed individuals. The proposed algorithm is evaluated based on empirical data on epidemy dynamics. Compared to sampling approaches, the novel model and its variants promise superior performance both in terms of efficiency and accuracy.

Qualitative Assessment

Overall, given the results, the paper presents a promising approach for evaluating social dynamics in large data sets as an alternative to sampling approaches. The paper is densely written, making it hard to follow at times. There are quite a number of grammatical errors/typos, which make reading the formal sections of the paper challenging, especially if not strongly acquainted with the area. A good example would have helped to guide the reader. Some typos that I noticed (I eventually stopped noting them down): - Line 55: `... in the large social systems.' (remove the) - Line 70: CHMM has not been introduced as an acronym - Line 72: `... defnied ...' - Line 98: `... probability of (an) event.' - Check paragraph starting in Line 103 for readability. - Line 107: Is `Variational Inference to ....' correct (Given the use in the remainder of the text, should it be perhaps be `for' or `with'?) - Line 196-198: Check sentence for correctness. - Line 200: `symptom' Missing `s'? - Line 229: `(explain the task later)' Is that a comment? - Table 1: `Poeple' --> People The evaluation exclusively concentrates on the comparison to sampling techniques (Gibbs, particle filtering). It would be worthwhile to compare it to other approaches introduced as part of the related work in order to get a more differentiated comparison. Moreover, how does the model perform if you pick larger or smaller subsets than the chosen 20% and 10% of the evaluated datasets? (In other words, why did you choose those numbers?) Overall, the paper makes interesting promises, but those should be better contextualised with the status quo of the field, and be presented in a clearer fashion.

Confidence in this Review

1-Less confident (might not have understood significant parts)


Reviewer 4

Summary

In this work the author propose the use of a stochastic kinetic model to track disease transmision for each individual of the grouop.

Qualitative Assessment

I really liked this paper. It is an interesting problem and the proposed solution seems that it is better than the previously proposed method with the advantage of lower computational time. The only small detail that I would point out is table 1, "Poeple" I guess is "People".

Confidence in this Review

1-Less confident (might not have understood significant parts)


Reviewer 5

Summary

In this paper, the authors first summarize and compare the coupled hidden Markov model and the stochastic kinetic model. Then they use the stochastic kinetic model, which successfully describes the interactions of molecules in chemical reactions, to characterize the social dynamic interactions and focus on the individual behaviors. This paper proposes the variational algorithm with stochastic kinetic model to make tractable inferences in social dynamics. The empirical results are shown by applying the proposed algorithm to the epidemic data along with the proximity data collected from sensor networks.

Qualitative Assessment

This paper introduces the stochastic kinetic model into social dynamics which is novel. I like the idea. This paper also shows the parameter learning method and infection predictions. The analysis is solid. Minor questions: - (line 102) You may need to give the definition of v_i first before you write down the equation here. I find that you give the definition in the later content (line 117). In addition, the variables used in equation 6 are not clear to me, i.e, line 132 to 133. - The real data set only contains 65 students. I think this data set is not large enough and may make the experimental results less convincing. Why do you choose this data set?

Confidence in this Review

1-Less confident (might not have understood significant parts)